# Do AutoML-Based QSAR Models Fulfill OECD Principles for Regulatory Assessment? A 5-HT_1A_ Receptor Case

**DOI:** 10.3390/pharmaceutics14071415

**Published:** 2022-07-06

**Authors:** Natalia Czub, Adam Pacławski, Jakub Szlęk, Aleksander Mendyk

**Affiliations:** Department of Pharmaceutical Technology and Biopharmaceutics, Jagiellonian University Medical College, Medyczna 9 St., 30-688 Kraków, Poland; natalia.czub@doctoral.uj.edu.pl (N.C.); j.szlek@uj.edu.pl (J.S.); aleksander.mendyk@uj.edu.pl (A.M.)

**Keywords:** 5-HT_1A_ receptor, AutoML, OECD principles, curated database, QSAR model

## Abstract

The drug discovery and development process requires a lot of time, financial, and workforce resources. Any reduction in these burdens might benefit all stakeholders in the healthcare domain, including patients, government, and companies. One of the critical stages in drug discovery is a selection of molecular structures with a strong affinity to a particular molecular target. The possible solution is the development of predictive models and their application in the screening process, but due to the complexity of the problem, simple and statistical models might not be sufficient for practical application. The manuscript presents the best-in-class predictive model for the serotonin 1A receptor affinity and its validation according to the Organization for Economic Co-operation and Development guidelines for regulatory purposes. The model was developed based on a database with close to 9500 molecules by using an automatic machine learning tool (AutoML). The model selection was conducted based on the Akaike information criterion value and 10-fold cross-validation routine, and later good predictive ability was confirmed with an additional external validation dataset with over 700 molecules. Moreover, the multi-start technique was applied to test if an automatic model development procedure results in reliable results.

## 1. Introduction

Quantitative structure–activity relationship (QSAR) models that have existed for many years are applied to the drug discovery process, offering in silico assessment and screening for the potential new drugs, which have not yet been assessed for in vitro or in vivo activity. Based on the molecule structure, it could be predicted whether a given compound will affect a chosen biological target and the strength of this effect [1]. The first successful QSAR model was developed in the 1960s by Hansch and Fujita [2] and concerned the toxicity of various chemicals on organisms. With the growing role of artificial intelligence (AI) in the drug discovery process, the quality of models and the scale of their application have increased [3].

Drug design and development is a challenging, costly, and time-consuming process and is simultaneously characterized by a high failure rate [4]. The discovery of new drugs is based on their efficacy in selected disease entities as well as their safety profile related to side effects after administration and drug–drug or drug–food interactions. The entire process, from the discovery of a new molecule to its introduction to the market, takes at least 10 years. The efficiency of this process is low [5]. Almost 90% of potential drugs fail between the first phase of clinical trials and the registration process [6]. Nowadays, artificial intelligence tools are used at various stages of the drug discovery and development processes [7]. Application of AI reduces the time needed to discover new drugs and the costs associated with in vitro and in vivo animal experiments. So far, AI does not replace laboratory experiments, but it is used as a complementary method. This is why it is so important to develop methods in the field of artificial intelligence [5]. They support the discovery of functions and structures of proteins. These methods allow us to discover drugs’ binding sites to the therapeutic targets [7]. Machine learning (ML), a field in artificial intelligence, is able to predict properties of compounds based on available data [8]. Using ML with a focus on a potential drug, various properties could be predicted: pharmacokinetic (rate of absorption [9] and elimination [10], volume of distribution [11], etc.), pharmacodynamic (ligand affinity/activity [12,13,14]), or the possible toxic effects [15]. Artificial intelligence can also estimate the likelihood of drug–drug interactions [16]. Another domain of AI application is chemical synthesis. Based on the data collected, AI can design a pathway for the synthesis of a new compound [17]. One of the many applications of AI in the search for new ligands is drug repositioning/repurposing, other therapeutic applications are sought among well-known drugs [18].

As the application of QSAR models became a routine in the process of development of new drugs, there was a need for regulations and guidelines to objectively determine the reliability of developed models. In 2007, the Organization for Economic Co-operation and Development (OECD) published “Guidance Document on the Validation of (Q)SAR Models”. This intergovernmental economic organization joins over 30 countries to coordinate, harmonize policies and work together to solve international issues [19]. The guidance is the work result of an expert group who started their work based on the “Setubal Principles”, which were first proposed at the international workshop in Setubal (Portugal) in March 2002 [20]. The OECD reduced the number of principles to five, which are presented below:A defined endpoint.An unambiguous algorithm.A defined domain of applicability.Appropriate measures of goodness-of-fit, robustness, and predictivity.A mechanistic interpretation, if possible [21].

The aim of this work was to create a QSAR model of ligand affinity for serotonin 5-HT_1A_ receptor according to the above principles. The presented work is a continuation of our research on the published preliminary model and the curated database [22]. To comply with all the OECD principles, we focused on reducing the number of descriptors while maintaining high-quality affinity predictions. The serotonin 5-HT_1A_ receptor belongs to the group of G protein-coupled receptors (GPCR) and is one of the best-studied receptors in the serotonergic system. It is mainly located in the brain (midbrain, limbic, and cortical regions) [23]. The serotonin receptor is an important biological target for researching new drugs in the field of central nervous system diseases [24]. Activation of 5-HT_1A_ is one of the mechanisms of action of antidepressants, anxiolytics, and antipsychotics [25]. Partial agonists (Aripiprazole, Clozapine, Buspirone, Trazodone, Ziprasidone) and 5-HT_1A_ receptor antagonists (Risperidone) are used in the treatment of depression, anxiety, schizophrenia, and bipolar disorder [26,27]. Currently, available medications have numerous side effects. For this reason, it is important to obtain a QSAR model that will effectively predict the affinity of potential molecules affecting this receptor. For this reason, it is important to obtain a QSAR model that will effectively predict the affinity of potential molecules affecting this receptor.

In the presented work, individual parts contain the database description (training and test sets), the results of experiments with the use of AutoML techniques, and the detailed analysis of the OECD principles in connection to this work.

## 2. Materials and Methods

### 2.1. Training Dataset

The curated database containing 9440 unique ligands of 5-HT_1A_ receptor, collected from ZINC and ChEMBL databases, was used in this study [28,29]. Curation of the database was described in previous research [22]. The affinity of compounds was presented by the negative logarithm of the constant inhibition, the pKi value. Affinity to 5-HT_1A_ receptor varied in the range between 4.2 and 11.0. Based on the Simplified Molecular Input Line Entry Specification (SMILES) of each molecule, Mordred 2D descriptors were obtained with Mordred package in Python 3 environment under a Linux operating system [30].

In our previous work, the number of inputs was reduced from over 1200 to 216 descriptors, which were then used to develop a preliminary model [22]. In this work, we use these 216 descriptors as the primary dataset.

### 2.2. Test Dataset

In order to evaluate the predictability of the model, an external dataset was prepared, which constitutes the test set. The data were retrieved from the GLASS database (https://zhanggroup.org/GLASS/, accessed on 1 September 2021). The GLASS (GPCR-Ligand Association) database is an experimental data repository on GPCR–ligand interactions. The sources of the data within the database are literature and public databases [31]. Originally, over 5000 ligands of the 5-HT_1A_ receptor affecting human cells were selected. The data cleaning stage included removing duplicates, compounds with affinity determined with a unit different than the inhibition constant (*K_i_*), and in the last step, the compounds that are also included in the curated database. Finally, 735 ligands were obtained with *K_i_* values, which were used to calculate pKi values.

### 2.3. Model

As in the case of the preliminary model, also in this study, the Automated Machine Learning (AutoML) tool was used to create predictive models. The H2O platform provides a tool to optimize the number of inputs (feature selection), algorithm selection, model development, and optimization of parameters [32]. The model was created using Python script, integrating both feature selection and 10-fold cross-validation (10-CV) schemes in the single non-interactive run [33].

### 2.4. Model Metrics

Four goodness-of-fit metrics were used to evaluate the developed models: root mean square error (*RMSE*), coefficient of determination (*R*^2^), *Adjusted R*^2^, and Akaike Information Criterion (*AIC*). Explanations of the metrics are presented below (Equations (1)–(4)). The performance of the QSAR model was assessed according to a 10-fold cross-validation (10-CV) scheme using the curated database. *Adjusted R*^2^ and *AIC* were applied to verify model performance on the test set. These values were important for obtaining information on how many inputs are valid for the QSAR model. *R*^2^ adjusted, similar to the coefficient of determination, should obtain the highest value. On the other hand, *AIC* and *RMSE* should be as low as possible.
(1)$$RMSE=\sqrt{\frac{\sum_{i=1}^{n}{\left(pred_{i}-obs_{i}\right)}^{2}}{n}}$$
where *RMSE* = root mean square error, *obs_i_* and *pred_i_* = observed and predicted values, *i* = data record number, and *n* = total number of records.
(2)$$R^{2}=1-\frac{SS_{res}}{SS_{tot}}=1-\frac{\sum_{i=1}^{n}{\left(pred_{i}-obs\right)}^{2}}{\sum_{i=1}^{n}{\left(obs_{i}-obs\right)}^{2}}$$
where *R*^2^ = the coefficient of determination, *SS_res_* = the sum of squares of the residual errors, *SS_tot_* = the total sum of the errors, *obs_i_* and *pred_i_* = observed and predicted values, and *obs* = arithmetical mean of observed values.
(3)$$Adjusted\;R^{2}=1-\frac{\left(1-R^{2}\right)\left(N-1\right)}{N-p-1}$$
where *R*^2^ = sample R-square, *N* = total sample size, and *p* = number of independent variables.
(4)AIC=2k−2ln(L)
where *AIC* = Akaike Information Criterion, *L* = likelihood function for the model, and *k* = number of estimated parameters.

The *AIC* value was calculated using the H2O Python module (H2O version = 3.32.1.6.) [34].

## 3. Results

### 3.1. Test Dataset

An introduction to the GLASS database is presented in the Materials and Methods (Section 2). It contains over 700 5-HT_1A_ receptor ligands. The pKi values ranged between 4.44 and 10.3, with a median of 7.22. In Figure 1, pKi’s distribution in GLASS (test set) and the curated database (training set) is presented. For both databases, the distributions are similar. The GLASS database was used for the evaluation of the QSAR model.

For a better comparison of databases, the following charts show the distributions of features representing Lipiński’s rule of five among drugs present in the GLASS and the curated database (Figure 2). The features’ distributions in both databases are similar, pointing out a good coverage of data space among training and external testing datasets.

### 3.2. Model

The AutoML tool produced several models with a reduced number of inputs. Table 1 shows top the five models with the number of selected descriptors and the original model based on 216 inputs. The *RMSE*, *R*^2^, *Adjusted R*^2^, and *AIC* were calculated according to the equations presented above.

All presented models above are stacked ensemble models, which consist of so-called base learners. The model development process in this case involves training a second-level “metalearner” to find the optimal combination of the base learners. In this work, we have applied GLM (generalized linear method) as a “metalearner” with a variable number of base models. It is observed that decreasing the input number causes an increase in the *RMSE* value, and a decrease in the determination coefficient in the case of the training set. This observation indicates the complexity of predicting the affinity value of the compound from its numerical representation towards the serotonin 5-HT_1A_ receptor.

The results for the external set show slightly worse results in comparison to the internal validation, which is expected. However, the differences are moderate, usually in the range 20% of the reference value. Overall, reported *RMSE* values indicate good predictability of the model also for compounds outside of the training set. In order to choose the best model, two additional measures of goodness-of-fit were introduced: *Adjusted R*^2^ and *AIC*. Both criteria take into account the model’s complexity and therefore in the case of similar predictions errors they favor simpler models. Based on the values of the *Adjusted R*^2^ and *AIC* on the test set, the optimal model is the one created based on 39 descriptors (*R*^2^*_adj_* = 0.5648, *AIC* = 1583.9). A fine consensus between model predictability and complexity was found escaping the curse of dimensionality and retaining high-level predictability of the QSAR model. As an additional diagnostic test for the model, the residual analysis was conducted, and a quantile–quantile plot (Q–Q plot) was drawn. It was observed that none of the prediction errors exceeded 25%, and for six molecules from close to 9500 in the database, the residual was between 20–25%. Moreover, based on the Q–Q plot it could be concluded that the predictions are normally distributed (Appendix A).

Table 2 shows the structure of the best model developed using 39 descriptors. AutoML H2O selected 17 models (from 340 initial), using GLM with Elastic Net module, and formed a second-level stacked ensemble model. 

As a point of reference, the linear model was established using Python package Linear Regression [35]. The linear model created on the input vector consisting of 39 input variables had the *RMSE* of 0.9718 and *R*^2^ of 0.1437, according to the 10-CV method.

### 3.3. Compliance with OECD Principles

The purpose of the OECD principles is to provide detailed information and guidelines explaining the application of validation rules to various types of QSAR models. Figure 3 shows principles from the Guidance Document on the Validation of (Quantitative) Structure-Activity Relationship Models and answers to how the QSAR model satisfies the rules, which is the subject of this paper.

#### 3.3.1. Defined Endpoint

The first OECD principle specifies that the created QSAR model should possess a clearly defined endpoint: a parameter that is predicted by a given model. The guidance specifies few groups of endpoints (physiochemical properties, environmental fate, ecological effects, human health effects) [21]. In our case, the endpoint is the pKi value, which is the negative logarithm of the inhibition constant (*K_i_*). It is an indication of how potent an inhibitor is and denotes the concentration required to produce half-maximum inhibition of the receptor (Equation (5)).
(5)$$P+I\overset{K_{i}}{⇌}P•I$$
where *P* = target protein, *I* = inhibitor, *K_i_* = inhibition constant, and *P*•*I* = the reversibly bound protein inhibitor complex [36]. 

There are many studies using the pKi value as an affinity value of compounds, in terms of affinity for serotonin receptors [37,38,39,40,41] or other biological targets [42,43,44,45,46]. Defined endpoint is a parameter that enables a comparison of its value between other studies. Without a doubt, pKi fulfills this requirement.

#### 3.3.2. Unambiguous Algorithm

The second principle, called “unambiguous algorithm,” specifies that the algorithm used to build the QSAR model should be precisely defined, ensuring transparency so that the others can re-create it and understand the model easily. The transparency of the model is based both on clearly defined descriptors building the QSAR model and modeling methods/techniques [21]. In our research, 2D chemical descriptors produced with the use of the Mordred package were applied. The authors of this package thoroughly described each descriptor [47]. This meticulous documentation and an Open Source code of the Mordred package ensures the clarity of the input information of the model, which fulfills part of this principle.

The second principle points out that transparency is also ensured by methods applied to develop the predictive model. According to the OECD documentation, an algorithm “may be a mathematical model or a knowledge-based rule developed by one or more experts”. The guidance presents algorithms commonly used in the QSAR modeling (Univariate regression, Multiple Linear Regression, Principal Component Analysis, Principal Component Regression, Partial Least Squares, Artificial Neural Nets, Fuzzy Clustering and Regression, K-nearest Neighbour Clustering, Genetic Algorithms) [21]. In our previous work, the obtained results were presented with the use of an automated machine learning technique in comparison with a simpler computational technique-linear regression model (LM) [22]. The LM structure and the principle of operation are easy to read and understand by a human. However, its simplicity leads to a much higher and unacceptable, from practical point of view, error value (*RMSE*) in comparison to a model developed by AutoML. These results indicate that simple techniques (readily interpretable and transparent) will not be able to create the model needed to predict the affinity of 5-HT_1A_ receptor ligands with high efficacy.

The AutoML methods have been used to create the 5-HT_1A_ receptor’s QSAR model. This technique reduces the transparency of the algorithm. To meet the second OECD rule, other prediction methods were sought, but the AutoML H2O model proved to be the best so far. Usage of AutoML techniques offers far greater possibilities than creating a simple QSAR model with, i.e., linear regression.

The stacked ensemble model based on 39 inputs is composed of several types of models, i.e., DeepLearning (seven models), GBM (three models), and XGBoost (seven models). Most of the modeling techniques mentioned above work under the clear principles, ensuring their transparency. However, artificial neural networks (ANN) including DeepLearning models are so called “black boxes”. The term is related to their complex structure and burden of computational operations beyond human perception capability. It is also associated with the elements of randomness included in the initialization of the model weights and the learning process by stochastic algorithms.

Compared to our preliminary AutoML model, which contained 342 models [22], the final QSAR model has fewer models (17) for better transparency. The authors of the OECD principles are particularly careful when, in the case of models based on neural networks, users must rely on a validation process to determine whether an ambiguous algorithm can produce reliable results in regulatory applications. For these reasons, we used an external database to test the resulting model.

The purpose of the second OECD principle, in addition to provide an explanation of how the model is produced, is the ability to be reproduced by other scientists. AutoML script is freely available to users [32] and the use of the same settings can lead to comparable results. Moreover, our final model with 39 inputs is freely available on the GitHub platform [48] and may be used for different data. For validation purposes, we conducted 30 independent experiments (multi-start method) with AutoML H2O script based on a curated database containing 39 descriptors and the same parameters (e.g., experiment duration, seed values), and we obtained highly reproducible predictions (Table 3). The results of one-way ANOVA for the training dataset were F value = 0.0002 and *p*-value = 1.0000 (0.9999999), respectively (Appendix A). The ANOVA results for the external dataset were F value = 0.0085 and *p*-value = 1.0000 (0.9999999) (Appendix A). For the second metric to determine the dispersion of predicted values, we examined the coefficient of variation (RSD) value. The use of a curated database did not exhibit high diversity, with respect to the predicted pKi values (RSD was in the range of 0.09–1.54% and the median was 0.28%). In the case of the GLASS database, the RSD value was in the range of 0.12–1.27%, and the median = 0.36%. As expected, these results show that the AutoML technique provides non-identical predictions, when started multiple times on the same data. However, the differences between results are not statistically significant. Moreover, all the final models produced by these 30 repeated tests are ensemble models and the types and number of models within their structures are also comparable.

The last part of the principle “unambiguous algorithm” demands explanation on how the QSAR estimates are obtained. This in fact leads directly to the fifth OECD principle mechanistic interpretation, which is included in the following sections of this article (Section 3.3.5). However, another angle regarding versioning of the software needs to be raised here. Open Source is based on the frequent publishing/updating policy, so the software versions are changing rapidly. AutoML by H2O is equipped with a strict version control system and reporting version of the modeling software at the beginning of the modeling procedure and when the binary object representing trained model is being opened. If any mismatch between the version of the saved model and the software used for its opening is being reported, it stops the model from being opened. Therefore, no unexpected behavior of the model is possible, and the software itself performs rigorous version checks.

#### 3.3.3. Applicability Domain

Another OECD principle is “defined field of application”. This rule sets the range of molecules for which the QSAR model should lead to correct predictions. The importance of this principle is that the model can be expected to provide reliable forecasts for chemicals that are similar to those used in model development [21]. Predictions that go beyond the applicability domain (AD) are extrapolations and are less likely to be reliable. The domain is limited to ligands affecting the 5-HT_1A_ receptor. 

The obtained model was created based on a diverse database in terms of structure. According to the Tanimoto coefficient, the curated database is characterized by the degree of similarity of the molecules to each other, in the range between 0.1553 and 1.0 (median = 0.37) [49]. The above results indicate that the ligands in the training set are highly diversified, which results in the possibility of using the QSAR model to predict compounds of various chemical structures. The model is not limited to a specific group of derivatives in terms of molecular structure. The limitation of AD is the pKi value, which for the assay set is in the range between 4.2 and 11, so if the model is used to predict the value of the compound’s affinity for the 5-HT_1A_ receptor, where the value will be outside this range, it may lead to poor or completely incorrect prediction of the pKi value.

The database from which the predictive model was created is characterized by diversity also in terms of parameters such as molar mass (149–1183, median = 406) or polar area (3.24–211.18, median = 55.87). The distributions of the values of the individual Lipinski’s rule parameters located at the point describing the comparison of the training and test databases are presented in Figure 2. These values indicate that the curated database contains the most compounds whose parameters indicate a high potential to become a drug. With respect to the applicability domain, the model is able to predict pKi values for substances that are potential drugs.

#### 3.3.4. Measures of Goodness-of-Fit, Robustness, and Predictivity

The fourth principle of the OECD Guidance focuses on the need to perform statistical validation of the QSAR model. This process is divided into the performance of the internal validation obtaining the goodness-of-fit and robustness of the algorithm, and the predictability obtained from the performance of the external validation. Due to the significant need to validate the model, the OECD organization defines in this principle appropriate measures related to the fit, robustness, and predictability to the created predictions of compounds affinity based on the structure [21]. 

The expert group listed in the document the most popular techniques for internal validation. They are cross-validation (leave-one-out (LOO) and leave-many-out (LMO)), bootstrapping, Y-scrambling or response permutation testing, and training/test set splitting [21]. The technique used in our study was ten-fold cross-validation (10-CV) as leave-many-out (LMO) procedure. Groups of compounds for each fold were chosen randomly and only used once for internal model validation.

External validation is the only way to determine the predictability of a QSAR model. The external set that has not been attached to the database used to create the model applies here, so the test set does not affect the development of the model. External validation does not replace internal validation, but only complements it. To make statistical conclusions about the predictability of the model compounds’ affinities against the 5-HT_1A_ receptor, the GLASS database (over 700 compounds), described in the Materials section (Section 2), was used. 

In this experiment, two parameters were used for the curated database (training set)-internal validation: *RMSE* and coefficient of determination. Additionally, these parameters were used for GLASS database (external validation) and two more parameters: *Adjusted R*^2^ and Akaike Information Criterion. *R*^2^ and *Adjusted R*^2^ are mentioned in the guidance as an example of a parameter determining a good fit of the model. On the other hand, the error value in the OECD document was presented, inter alia, as mean squared error (MSE). In our work, we use the square root of this value (*RMSE*).

#### 3.3.5. Mechanistic Interpretation

Mechanistic interpretation is not a mandatory principle to be fulfilled. However, an explanation of the relationship between the chemical structure and affinity may confirm the acquired knowledge and expand it on these dependencies. The fifth OECD principle encourages the validation process to find mechanistic interpretations that can contribute to a better understanding of statistical validity and the applicability domain. Compounds in these models are presented as molecular descriptors, a mathematical representation of the structural features of molecules [21].

It is important that the method of calculating molecular descriptors is accessible to the user and that it can be applied reproducibly to all chemical structures. Therefore, in our research, 2D chemical descriptors produced by the Mordred package were used. All features were described by the authors, which gives the possibility of model interpretation [47].

AutoML techniques are proficient in selecting descriptors relevant to the model. Not all descriptors may be important for prediction, that is why feature selection is an important step of model development. The OECD guideline points out that a large number of variables can cause modeling errors due to the overfitting of model to data, which reduces its robustness and generalization ability. Therefore, the reduction in the parameters presented here by the numerical representation of the molecules leads to a more general model that allows affinity prediction for compounds, not in the curated database on which the model was created. In our previous work, we showed mechanistic interpretation of model using SHAP analysis [22].

##### Shapley Additive Explanations (SHAP)

The best model was analyzed in order to elucidate complex interrelations between input variables and predicted pKi values using Shapley Additive Explanations (SHAP) method. SHAP analysis provides an objective assessment of the input of every single variable on the final outcome of the model. SHAP calculations are based on the theory of cooperative games provided by Lloyd Shapley in 1951 and resulted in a Nobel Memorial Prize in Economic Sciences in 2012. The theory allows the assignment to every cooperative game a unique distribution among all players of a total surplus generated by the coalition of all players. Shapley values were found to be useful in the explanation of predictive models, especially within the machine learning domain. Based on that, the model’s inputs are treated as players and the predicted value is the outcome of the coalition. The Shapley value provides an answer to the question of how important the input of each player is (variable in our case) to the overall result (prediction), and what share can be assigned to it. The contribution of every participant should be proportional to their marginal contribution in-game outcome. The Shapley value allows the assessment of an individual’s contribution to the final results in an efficient, symmetrical, and additive way, including the possibility to detect inputs with zero contributions. The marginal contribution for each individual is calculated by generating all permutations of individuals and their results obtained by the formed coalition. The result from the computational analysis provides a ranking of variables with absolute average SHAP value. Moreover, there is the possibility of assessing the influence of variable value on the final model outcome. It might be positive or negative, and its magnitude may differ within a variable range [50,51].

The SHAP plots were produced using an in-house developed Model Interpretation tool, which is publicly available on the GitHub website [52]. The overall variable importance might be reflected by the mean absolute SHAP value which indicates how much on average the variable affects the predicted pKi value. By investigating the obtained results, it might be observed that the marginal contribution for every variable differs and all features are in the range of 0.021–0.088. A short summary of the calculations for the 10 most important variables is presented in Table 4, whereas full results including variable descriptions are available in Appendix A.

The two most important variables are characterized by mean absolute SHAP value equal to 0.088: MOE MR VSA Descriptor 3 (SMR VSA3) and Geary autocorrelation of lag 3 weighted by polarizability (GATS3p). Absolute value of SHAP above 0.08 was also calculated for the following three variables: PEOE VSA2, SaaaC, and AATSC3. Moreover, it is observed that variable contribution to prediction might be positive or negative in relation to its value itself. By taking the five mentioned above, a high value of SMR VSA 3 and GATS3p negatively impacts the pKi predicted by the model, whereas a higher value of PEOE VSA2, SaaaC, and AATSC3se positively influences the pKi predicted by the model (Figure 4).

Model analysis in a more detailed way shows trends between variables and their impact on the predicted outcome by the model. For example, Figure 5 presents the relation of the calculated SHAP value to SMR VSA3. It is observed that a value equal to or higher than 15 causes a drop in pKi predicted by the model. In the case of SMR VSA3 ≤ 10, it may be expected that compound affinity for 5-HT_1A_ receptor will increase. SMR VSA represents the polarizability of a molecule; therefore, in light of the observed relation between the descriptor and SHAP value, it might be assumed that too many polarizable groups are not desired in the case of active compounds. Different MR VSA descriptors were found to be important to predict the affinity of compounds against molecular targets. An example might be the model of human ether-a-go-go-related gene (hERG) blockers developed by Moorthy et al. [53].

A closer look into the impact of GATS3p on predicted pKi shows that a descriptor value below 1.0 indicates a higher affinity of a molecule to the 5-HT_1A_ receptor (Figure 6). It is worth pointing out that GATS3p has also embedded information about the polarizability of the compound. Topological distribution of polarizability was identified as important for predicting the affinity of molecules against other targets such as mitogen-activated protein kinase-interacting kinases (MNK1, MNK2) [54] or apoptosis inducers for human breast cancer cell line T47D and human colorectal cancer cell line DLD-1 [55].

Analysis of the variables’ impact on the pKi value predicted by the model could also be directed from global effects and condensed into local and more detailed form. Going from SHAP analysis to other available methods of model explanation, it might be advisable to apply the partial dependence plot (PDP) method that shows the marginal effect of one or two features on the predicted outcome by the model [56,57]. PDP can show whether the relationship between predictions and input variable is linear or more complex. For example, when applied to a linear regression model, PDP shows a pure linear relationship, whereas for neural networks, the expected relation will be nonlinear and more complex. PDP analysis is implemented in the AutoML H2O package and developed models can be easily analyzed using internal functions delivered by software. PDP analysis was conducted using an in-house developed tool built on the H2O package, capable of automatically exporting results in the graphical form of 3D plots [52]. An example of the PDP result for the two most important variables according to SHAP analysis (SMR VSA3 and GATS3p) is visualized in Figure 7. It is observed that in the case of both variables, lower values positively influence the pKi predicted by the model. The opposite direction in model outcome is observed for higher values of both variables. It is worth mentioning that within the analyzed domains, the functional relationship is not monotonic. Due to the nonlinearities affecting the predictions for both variables, the result of the interaction is complex. It might be challenging to describe it by a single number representing directional effect such as in linear models with variable interaction terms. Such simplification will be possible, but it will be occupied by the loss of information represented by the model. The complete result of PDP analysis, depicted by over 700 unique plots, is provided as Appendix A.

The model’s explanation could be conducted in more detailed way in order to find local effects using Individual Conditional Expectation (ICE) or Local Interpretable Model-agnostic Explanations (LIME) methods. ICE method focuses on individual data instances and provides the functional relationship between predictions for a single record by changing the chosen feature value. ICE plot visualizes the dependence of the model’s prediction and the feature for each instance separately, resulting in a single line per data record, compared to one line per feature in one-dimensional PDP. The result of the analysis is a set of points showing changes in single instance outcome in relation to feature modification [58]. ICE for a single feature can be computed by keeping all other variables unchanged. In the case of QSAR models, that might be helpful when the single structure is an object for further modifications. It could also help with a better understanding of the impact of changes in selected features on a compound’s expected activity or properties. On the other hand, the LIME method relies on new dataset generation that consists of perturbed samples from the original database and using the model under analysis to generate predictions. Then, an interpretable model, such as linear regression or decision tree, is trained, weighting the proximity of the sampled instances to the instance of interest. The learned model should be a good approximation of the machine learning model predictions locally, but it does not have to be a good global estimate. LIME can also lead to the misinterpretation of results due to the possibility of unlikely values of features in generated database utilized for surrogate model development.

## 4. Discussion

Based on a conducted literature review, no work has been published that includes a QSAR model predicting the affinity of ligands against the 5-HT_1A_ receptor while meeting the OECD guideline for (Q)SAR model validation—this also includes our preliminary findings, where no OECD angle was taken into consideration [22]. There are several models that meet OECD regulations concentrated on the other biological targets: HIV-I reverse transcriptase, tyrosinase, dopamine transporter, Mer kinase, and SIRT1. In these examples, multiple linear regression, evolutionary computation, and Monte-Carlo-based methods were applied to develop QSAR models. The above-mentioned research examples were mainly conducted on the databases of about 50 compounds, with one exceeding the number of 100 molecules, and just one database including more than 1400 compounds [59,60,61,62,63]. Our project is an extension of previous works both in terms of the choice of biological target (5-HT_1A_) and the size/diversity of the training database (almost 9500 unique compounds). A large number of molecules in a database and their structural and physicochemical diversity cause difficulties in the development of the QSAR model using simple modeling tools together with the satisfying quality of predictions.

In this paper, besides the development of a predictive model, we provided a detailed explanation of whether our model satisfies the OECD rules. Our model fully satisfies four of the five OECD principles (defined endpoint; defined domain of applicability; appropriate measures of goodness-of-fit, robustness, and predictivity; mechanistic interpretation). The only principle partially satisfied is the ‘second principle’—unambiguous algorithm. The Mordred descriptors used to create the model are clearly and thoroughly documented by the software developers, which is the source of transparency of the resulting QSAR model. OECD recommendations indicate that the algorithm must provide information on how accurately the estimated values were obtained so that other scientists can reproduce the calculations. Our final model, based on 39 inputs, has a structure of expert committee in which artificial neural networks are present among other algorithms. This introduces ambiguity into the predictions obtained. The guideline [21] for the implementation of a neural network-based model indicates the need for external validation of the model. In our case, external validation was performed using the GLASS database [31].

In order to prove the reproducibility of the model development process, we performed 30 independent attempts to create the model, setting the same parameters using the AutoML tool. The numerical values of predictions in both testing and validation procedures were almost the same for every repetition. As our experiments showed, it was impossible to obtain a good model with a low error and high coefficient of determination without using complex computational tools such as deep learning methods. However, in our study, we showed that multiple experiments under the same conditions using the AutoML H2O tool led to almost the same results. The differences between predictions were found to be statistically insignificant. For 8 out of nearly 9500 compounds, the RSD (relative standard deviation) value exceeded 1%, whereas the highest RSD value of 1.54% (curated database) was observed for all 30 repetitions. When converted to *Ki* (nM) values, which is the primary outcome in receptor affinity tests, the RSD in the predicted values for this compound is 19.33%. Food and Drug Administration regulations—“Bioanalytical Method Validation Guidance for Industry”—on ligand binding assays indicate a level of precision between assays of ±20%. Our predictions’ diversity value falls within this range. The information provided indicates that our final model leads to affinity predictions that are within the error range indicated by the FDA [64].

## Figures and Tables

**Figure 1 pharmaceutics-14-01415-f001:**
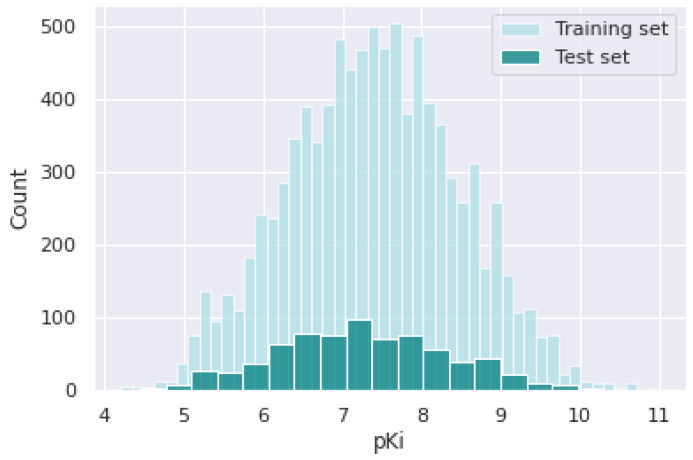
Histogram of pKi value in training and test sets.

**Figure 2 pharmaceutics-14-01415-f002:**
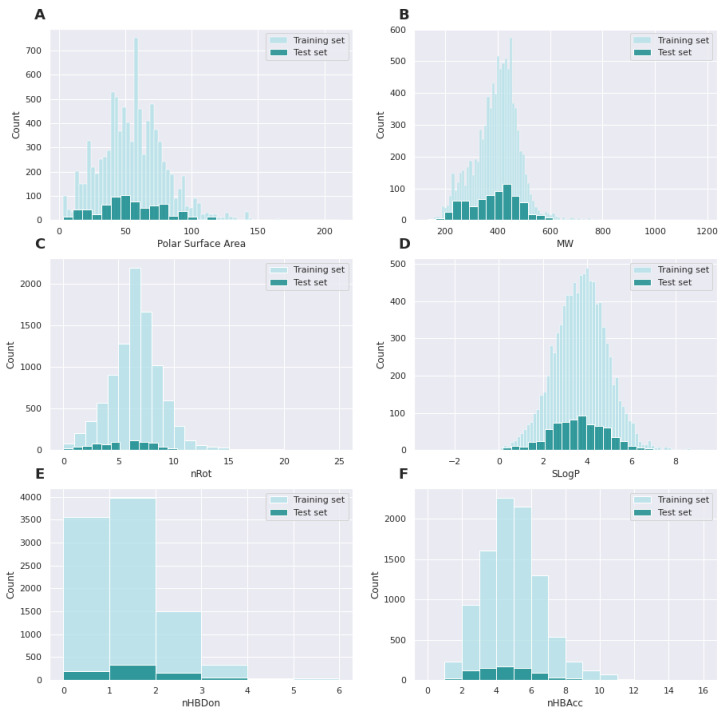
Distribution of Lipinski’s rules features values in training and test sets. (**A**) Polar surface area; (**B**) MW, molecular weight; (**C**) nRot, rotatable bonds; (**D**) SLogP, logP value; (**E**) nHBDon, number of H-bonds donors; (**F**) nHBAcc, number of H-bond acceptors.

**Figure 3 pharmaceutics-14-01415-f003:**
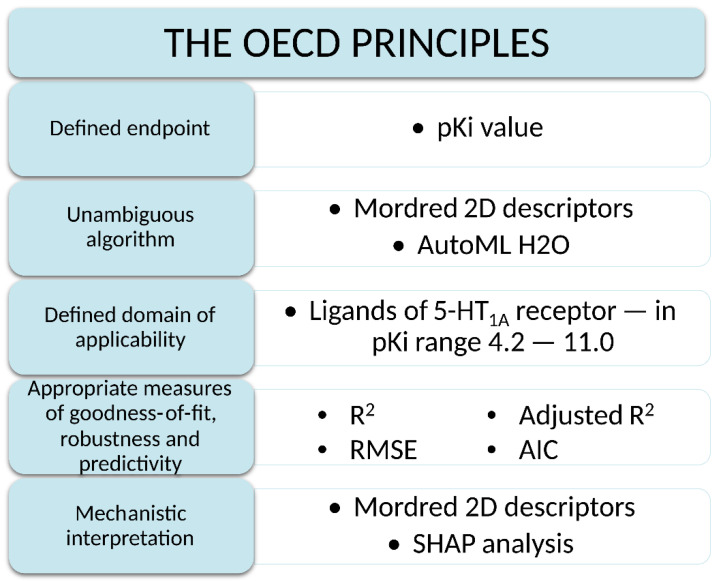
Scheme of the OECD principles.

**Figure 4 pharmaceutics-14-01415-f004:**
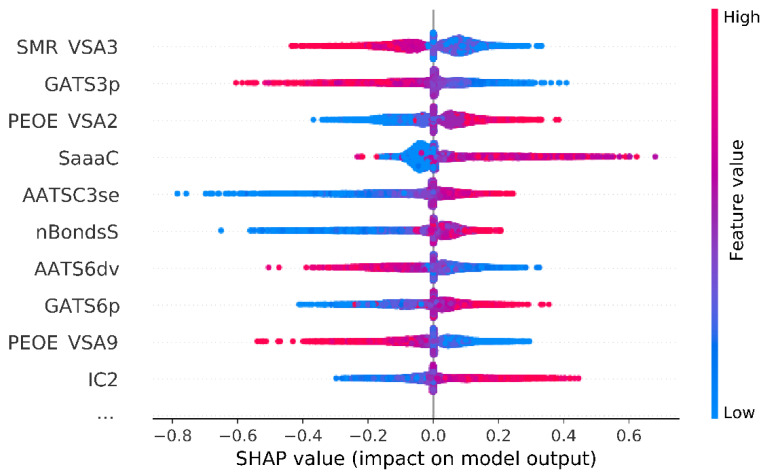
Summary of SHAP analysis for 10 input variables with the overall highest impact on model predictions. SMR VSA3—MOE MR VSA Descriptor 3; GATS3p—Geary autocorrelation of lag 3 weighted by polarizability; PEOE VSA2—MOE Charge VSA Descriptor 2; SaaaC—sum of aaaC; AATSC3se—averaged and centered Moreau–Broto autocorrelation of lag 3 weighted by Sanderson EN; nBondsS—number of single bonds in non-kekulized structure; AATS6dv—averaged Moreau–Broto autocorrelation of lag 6 weighted by valence electrons; GATS6p—Geary coefficient of lag 6 weighted by polarizability; PEOE VSA9—MOE Charge VSA Descriptor 9; IC2—2-ordered neighborhood information content.

**Figure 5 pharmaceutics-14-01415-f005:**
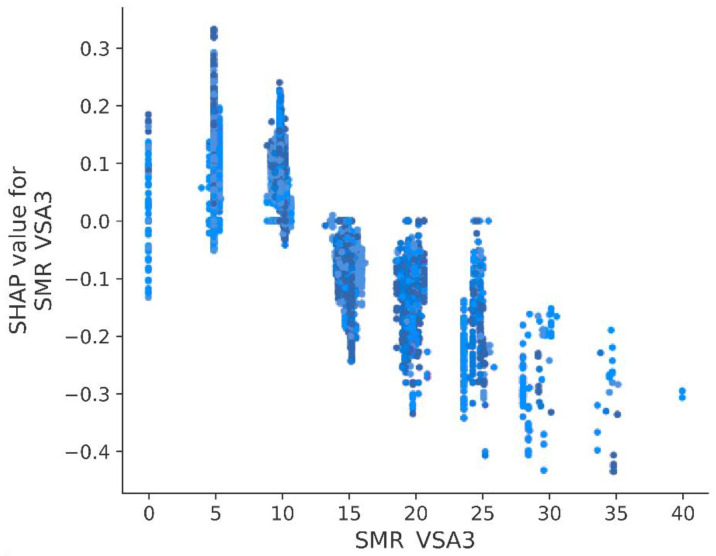
Calculated SHAP value in relation to MOE MR VSA Descriptor 3 (SMR VSA3).

**Figure 6 pharmaceutics-14-01415-f006:**
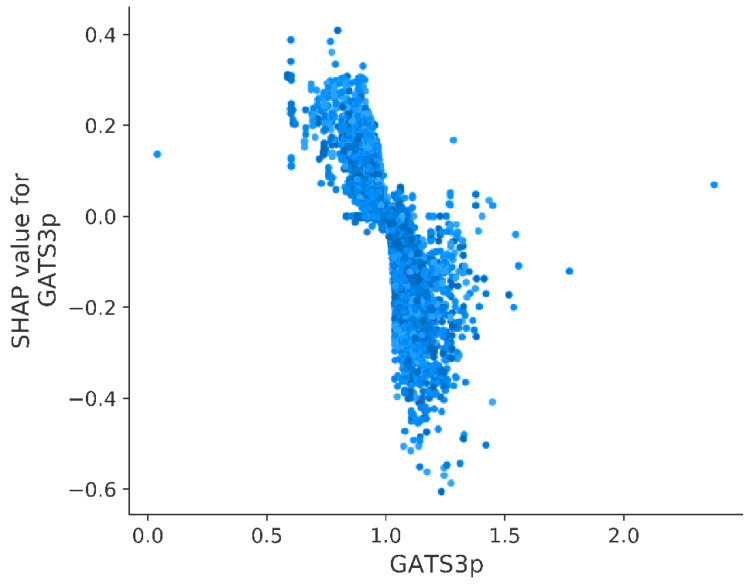
Calculated SHAP value in relation to Geary autocorrelation of lag 3 weighted by polarizability (GATS3p).

**Figure 7 pharmaceutics-14-01415-f007:**
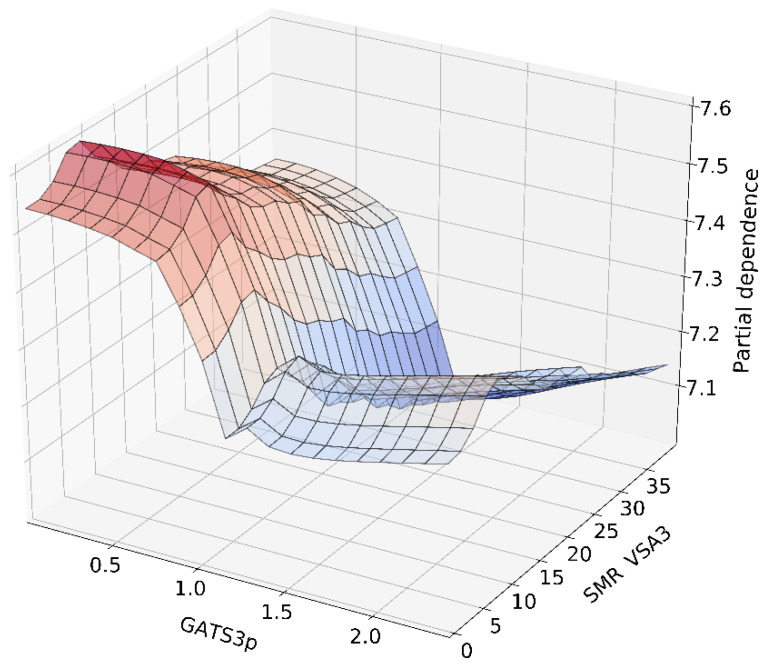
Two-dimensional partial dependence plot visualizing the interaction and effect on the pKi value precited by the model for the two most important variables based on SHAP analysis: SMR VSA3 and GATS3p.

**Table 1 pharmaceutics-14-01415-t001:** Results of the AutoML model evaluation for training and test sets. *RMSE*—root mean square error; *R*^2^—coefficient of determination; *AIC*—Akaike Information Criterion.

Inputs Number	10-CV		External Testing			
	*RMSE*	*R* ^2^	*RMSE*	*R* ^2^	*Adjusted R* ^2^	*AIC*
216	0.5437	0.7443	0.6806	0.6021	0.4362	1642.8
123	0.5523	0.7361	0.6830	0.5992	0.5185	1605.5
**39**	**0.5774**	**0.7116**	**0.6926**	**0.5879**	**0.5648**	**1583.9**
38	0.5782	0.7108	0.7282	0.5445	0.5196	1673.5
24	0.5926	0.6962	0.7276	0.5452	0.5298	1716.4
23	0.5941	0.6946	0.7597	0.5042	0.4882	1751.8

**Table 2 pharmaceutics-14-01415-t002:** The structure of stacked ensemble with the coefficients of GLM with Elastic Net as a “metalearner” model (39 inputs).

Name	Coefficients
Intercept	−1.0046
GBM_grid__1_AutoML_20210902_051708_model_54	0.6633
GBM_grid__1_AutoML_20210902_051708_model_20	0.1599
DeepLearning_grid__3_AutoML_20210902_051708_model_3	0.1089
XGBoost_grid__1_AutoML_20210902_051708_model_120	0.0848
DeepLearning_grid__3_AutoML_20210902_051708_model_8	0.0295
DeepLearning_grid__2_AutoML_20210902_051708_model_2	0.0263
DeepLearning_grid__3_AutoML_20210902_051708_model_2	0.0187
DeepLearning_grid__3_AutoML_20210902_051708_model_5	0.0058
XGBoost_grid__1_AutoML_20210902_051708_model_52	0.0066
DeepLearning_grid__2_AutoML_20210902_051708_model_3	0.0074
XGBoost_grid__1_AutoML_20210902_051708_model_95	0.0058
XGBoost_grid__1_AutoML_20210902_051708_model_131	0.0058
XGBoost_grid__1_AutoML_20210902_051708_model_90	0.0037
XGBoost_grid__1_AutoML_20210902_051708_model_113	0.0026
DeepLearning_grid__2_AutoML_20210902_051708_model_8	0.0025
XGBoost_grid__1_AutoML_20210902_051708_model_30	0.0026
GBM_grid__1_AutoML_20210902_044957_model_20	0.0005

**Table 3 pharmaceutics-14-01415-t003:** Results of one-way ANOVA tests from the multi-start method (30 repetitions).

Measure	Curated Database	GLASS Database
F value	0.0002	0.0085
*p*-value	1.0000	1.0000
Statistically significant different predictions	False	False

**Table 4 pharmaceutics-14-01415-t004:** Summary representing mean absolute SHAP value (av|SHAP|) for the 10 most important variables.

Variable	av|SHAP|	Description
SMR VSA3	0.088	MOE MR VSA Descriptor 3
GATS3p	0.088	Geary autocorrelation of lag 3 weighted by polarizability
PEOE VSA2	0.083	MOE Charge VSA Descriptor 2
SaaaC	0.083	Sum of aaaC
AATSC3se	0.082	Averaged and centered Moreau–Broto autocorrelation of lag 3 weighted by Sanderson EN
nBondsS	0.078	Number of single bonds in non-kekulized structure
AATS6dv	0.073	Averaged Moreau–Broto autocorrelation of lag 6 weighted by valence electrons
GATS6p	0.071	Geary coefficient of lag 6 weighted by polarizability
PEOE VSA9	0.071	MOE Charge VSA Descriptor 9
IC2	0.069	2-ordered neighborhood information content

## Data Availability

Final model file based on 39 descriptors, stacked ensemble model compatible with H2O v. 3.32.1.6. Available online: https://github.com/nczub/5-HT1A_affinity_prediction_model/tree/main/39in_AutoML_H2O_model (accessed on 6 May 2022).

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
