# Peer review of "Do AutoML-Based QSAR Models Fulfill OECD Principles for Regulatory Assessment? A 5-HT1A Receptor Case"

_pharmaceutics, 2022, doi:10.3390/pharmaceutics14071415_

Round 1

Reviewer 1 Report

PacÅ‚awski and wo-corkers report on a predictive model for the serotonin 1A receptor affinity and  the explanation in detail of its validation according the OECD rules. The manuscript constitutes an interesting contribution to the field, it is easy to read, it is critical, and the compilance of OECD rules were competently analyzed.

Author Response

Reviewer 1: 

PacÅ‚awski and wo-corkers report on a predictive model for the serotonin 1A receptor affinity and  the explanation in detail of its validation according the OECD rules. The manuscript constitutes an interesting contribution to the field, it is easy to read, it is critical, and the compilance of OECD rules were competently analyzed. 

Answer: 
We would like to thank you for your kind words, and we appreciate that the presented results were found to be interesting. We also believe that our findings as well as presented model will be a good starting point for future discussion in the field of QSAR modeling. 

Reviewer 2 Report

This is an interesting discussion applying AutoML in the context of OECD principles. The authors have a quite thorough discussion on how the principles can be satisfied using the AutoML approach. The article could be published after the authors provide more details and make their argument more solid. It would be ideal to publish the article with complete and self-contained discussions to give readers of Pharmaceutics most value.

Comments

  1. Line 36 - 38: The authors claimed AI can be used to predict PK, PD, or toxic effects. Is it possible to add a reference to support this claim? Same for the rest of this paragraph — It would be helpful to the readers to know what are the references to look for and the authors need to justify these statements.
  2. Part 3.1 and Table 1: Model results
    1. Details of the models generated by AutoML should be mentioned here. What types of models are included and used in AutoML training? And what are the types of the models listed in Table 1? Are they the same typed models or different?
    2. Metrics of training data and test data are used. However, it is not clear whether the RMSE and R2 values of training data are from 10-CV or just the final model on training?
    3. Figure 3: The authors need to justify why AutoML can be considered as “unambiguous algorithm”. Without explanation and details, it is not convincing AutoML is unambiguous since the open-sources libraries are constantly evolving so the results will not be the same and not reproducible unless it is stored in some immutable versioning system. The authors need to clearly layout the model operation and versioning procedure.
    4. It is necessary to provide more details on model performance to give readers better idea of how much they can trust the model and what might be the pitfalls. For example, actual v.s. predicted, Q-Q plot of the residual, residual as a function of predicted / other variables, in addition to just RMSE, R2, and AIC. Is there any outlier or special case that the model gives very bad predictions?
  3. Line 230: More elaboration is needed to describe the stacked model — How are the model stacked together? What is the hierarchy for model stacking? It would be nice to visualize a simple stacked model to give readers better idea.
  4. Line 233-235: I do not agree that the reason of “black box” model is because of the randomness or stochastic algorithms.
  5. Line 236: What is the preliminary AutoML model that has 342 models? Is it possible to point out where this has been described somewhere in the paragraphs?
  6. Figure 4 - 6: It’s only for aesthetic reason but is it possible to change the background of these figures to white such that it is consistent with the background and not obvious it is a cropped screenshot?
  7. Line 343: It is good to use SHAP as a method to get interpretability of the AutoML stacked models. A citation should be included in addition to the GitHub link: Lundberg, S. M., & Lee, S. I. (2017). A unified approach to interpreting model predictions. Advances in neural information processing systems, 30.
  8. It would be ideal to discuss more on different interpretable ML approached in addition to SHAP, such as ICE, LIME, etc. as well as some more references for readers who are interested in the topic to refer to. Also, it seems SHAP only gives an average effect from a single input variable. Is it possible for the authors to discuss more on how the factors interact with each other and how to interpret those higher-order effects (just like the cross terms in a polynomial model)?
  9. Line 498: Unfortunately the link to reference 7 (OECD Principles) is not working. Please replace it with the updated link.

Reviewer 3 Report

Comment:

The manuscript is very interesting, carefully prepared and covers important research. In the study the QSAR model predicting the affinity of ligands against the 5-HT1A receptor while meeting the OECD guideline (for (Q)SAR model validation ) was developed.

The authors pointed to the incomplete fulfillment of one of the OECD principles (the principle of "unambiguous algorithm"), which they linked with the structure of the derived model and supplemented their work with the external validation. The analysis of the obtained results covers all the features of a "good" QSARs model. The reproducibility of the model development process was also proved.

Comment:

The manuscript is very interesting, carefully prepared and covers important research. In the study the QSAR model predicting the affinity of ligands against the 5-HT1A receptor while meeting the OECD guideline (for (Q)SAR model validation ) was developed.

The authors pointed to the incomplete fulfillment of one of the OECD principles (the principle of "unambiguous algorithm"), which they linked with the structure of the derived model and supplemented their work with the external validation. The analysis of the obtained results covers all the features of a "good" QSARs model. The reproducibility of the model development process was also proved.

Author Response

Answer: 
We would like to thank you for your kind comment. We appreciate that the presented results were found to be interesting for the Reviewer. 

Reviewer 4 Report

Abstract. Results & Conclusion are poor, improve it because these two portions are an important part of this section 

Introduction. At-least one comprehensive paragraph is required about the drug development & Designing 

Introduction. 5-HT1A receptor, explain why authors need to design and explore new ligands for this receptor, and explain the drawbacks of former findings to give strength to this presented work 

Test dataset. This section only describes the methodology, the findings shall be in results, that is, pKi distribution, etc.

Supplementary. Some of the data must be incorporated into the main manuscript to understand the findings on one page 

Author Response

Abstract. Results & Conclusion are poor, improve it because these two portions are an important part of this section  

Answer: 
Thank you for this remark. We introduced additional information in the results section by adding a paragraph and a table with the final model structure description and discussing more in detail options for model explanation methods. There is also additional Supplementary material presenting PDP for 2-way variables interactions.  

Introduction. At-least one comprehensive paragraph is required about the drug development & Designing  

Answer: 
Thank you for this comment. An additional paragraph describing the R&D process was added. Lines:  34-48. 

Introduction. 5-HT1A receptor, explain why authors need to design and explore new ligands for this receptor, and explain the drawbacks of former findings to give strength to this presented work  

Answer: 
Thank you for this remark. An additional paragraph pointing need for further drug search acting by 5-HT1A receptor was added: lines 76-86 

Test dataset. This section only describes the methodology, the findings shall be in results, that is, pKi distribution, etc. 

Answer: 
Thank you for this suggestion. We moved all results from the methods section into the results – recently, lines 158-174 and Figures 1 and 2. 

Supplementary. Some of the data must be incorporated into the main manuscript to understand the findings on one page 

Answer: 
Thank you for this suggestion. Some additional diagnostics and explanations of the model were incorporated into the main text. However, we need to keep the main text simple enough to be comprehensible. AutoML models are usually complicated ensembles and that is why we “delegate” some of the more elaborate depictions of the model to the supplementary material. Additional depictions were provided by the summary of residuals analysis and Q-Q plot for the model. There is also a sample of a partial dependency plot (PDP) included in the results. Moreover, the structure of the final stacked ensemble is described in additional Table 2.